# An Updated Review on EPR-Based Solid Tumor Targeting Nanocarriers for Cancer Treatment

**DOI:** 10.3390/cancers14122868

**Published:** 2022-06-10

**Authors:** Majid Sharifi, William C. Cho, Asal Ansariesfahani, Rahil Tarharoudi, Hedyeh Malekisarvar, Soyar Sari, Samir Haj Bloukh, Zehra Edis, Mohamadreza Amin, Jason P. Gleghorn, Timo L. M. ten Hagen, Mojtaba Falahati

**Affiliations:** 1Student Research Committee, School of Medicine, Shahroud University of Medical Sciences, Shahroud 3614773947, Iran; majidsharifi@tabrizu.ac.ir; 2Department of Tissue Engineering, School of Medicine, Shahroud University of Medical Sciences, Shahroud 3614773947, Iran; 3Department of Clinical Oncology, Queen Elizabeth Hospital, Hong Kong, China; chocs@ha.org.hk; 4Department of Cellular and Molecular Biology, Faculty of Advanced Science and Technology, Tehran Medical Sciences, Islamic Azad University, Tehran 1916893813, Iran; ansariasal1379@gmail.com (A.A.); rahil_tarharoudi@yahoo.com (R.T.); hedyehmalekisarvar@gmail.com (H.M.); sari.s@iaups.ac.ir (S.S.); 5Department of Clinical Sciences, College of Pharmacy and Health Sciences, Ajman University, Ajman P.O. Box 346, United Arab Emirates; s.bloukh@ajman.ac.ae; 6Centre of Medical and Bio-allied Health Sciences Research, Ajman University, Ajman P.O. Box 346, United Arab Emirates; z.edis@ajman.ac.ae; 7Department of Pharmaceutical Sciences, College of Pharmacy and Health Sciences, Ajman University, Ajman P.O. Box 346, United Arab Emirates; 8Laboratory Experimental Oncology and Nanomedicine Innovation Center Erasmus, Department of Pathology, Erasmus MC, 3015 GD Rotterdam, The Netherlands; m.amin@erasmusmc.nl (M.A.); m.falahati@erasmusmc.nl (M.F.); 9Department of Biomedical Engineering, University of Delaware, Newark, DE 19713, USA

**Keywords:** enhanced permeability and retention (EPR) effect, nanocarriers, solid tumor, drug delivery, vascular diffusion

## Abstract

**Simple Summary:**

One of the important efforts in the treatment of cancers is to achieve targeted drug delivery by nanocarriers to be more effective and reduce adverse effects. However, due to the adverse responses of nanocarriers in clinical trials due to the very weak EPR effects, doubts have been raised in this regard. In this study, an attempt has been made to take a critical look at EPR approaches to enable the convergence of previous papers and the EPR critics to reach an appropriate therapeutic path. Although the effectiveness of EPR is highly variable due to the complex microenvironment of the tumor, there is high hope for cancer treatment by describing new strategies to overcome the challenges of EPR effect. Furthermore, in this paper an attempt was made to provide a reliable path for future to develop cancer therapeutics based on EPR effect.

**Abstract:**

The enhanced permeability and retention (EPR) effect in cancer treatment is one of the key mechanisms that enables drug accumulation at the tumor site. However, despite a plethora of virus/inorganic/organic-based nanocarriers designed to rely on the EPR effect to effectively target tumors, most have failed in the clinic. It seems that the non-compliance of research activities with clinical trials, goals unrelated to the EPR effect, and lack of awareness of the impact of solid tumor structure and interactions on the performance of drug nanocarriers have intensified this dissatisfaction. As such, the asymmetric growth and structural complexity of solid tumors, physicochemical properties of drug nanocarriers, EPR analytical combination tools, and EPR description goals should be considered to improve EPR-based cancer therapeutics. This review provides valuable insights into the limitations of the EPR effect in therapeutic efficacy and reports crucial perspectives on how the EPR effect can be modulated to improve the therapeutic effects of nanomedicine.

## 1. Introduction

Despite various approaches to control solid tumor growth, such as surgery, chemotherapy, radiation therapy, and thermotherapy, solid tumors are still the leading cause of death in cancer patients [1]. The need for better clinical outcomes in patients with cancer has led researchers to reconsider therapeutic strategies. To overcome these issues, the structural characteristics of solid tumors, drug nanocarrier transport in tumor vessels and the interstitium, selective drug delivery systems, and analytical tools should be carefully considered to develop cancer therapeutics [2,3]. Although it has long been thought that drug nanocarriers will facilitate cancer therapeutics by directing drugs to solid tumors, nanomedicine development has not yet achieved promising clinical outcomes. Numerous reports indicate that virus-/inorganic nanoparticles-/organic nanoparticles-based nanocarriers could improve the efficacy and safety of anticancer drugs through potential targeting, mitigated drug release in non-target tissues, and pH-sensitive drug release in the tumor microenvironment (TME) [4,5,6,7]. Tumor vascular permeability and the enhanced permeability and retention (EPR) mechanism in macromolecular drug delivery to solid tumors have been reported as successful strategies for cancer therapeutics [8,9,10]. However, the low EPR effect in human solid tumors [8], poor tumor extravasation [11] and infiltration [11], and a diversity in the tumor-immune microenvironment (TIME) [12] has resulted in the full realization of EPR-mediated cancer therapeutics [13]. 

Today, simulation models have reported the dynamics of tumor vessel cooption and have promoted the treatment of solid tumors [14,15]. However, different results were obtained from different strategies aimed at investigating EPR-mediated drug delivery to cancer cells. Different results can be related to the animal species used, the extent of intratumoral ECM content, an unorganized tumor vasculature, the diverse properties of drug nanocarriers, variable fluid velocities, the types of malignant cells in the TIME despite having the exact origin, and the heterogeneous tissues of a solid tumor [16,17,18,19,20,21]. Despite the certainty of the EPR-mediated tumor targeting and cancer nanomedicine treatment efficacy [22], the negative side of the EPR effect in cancer nano-therapy is derived from the heterogeneous microvascular networks of solid tumors, which results in heterogeneous distributions of nanomedicines. Therefore, a full exploration of the EPR effect in solid tumors and issues associated with dynamics, such as pathophysiological and pathoanatomic characteristics, drug nanocarrier formulations, physicochemical factors, and EPR analytical methodology, can provide useful information in the development of EPR-based therapeutics.

In this regard, Nichols and Bae [9] discussed that the EPR approach in the therapeutic activities of animal tumors is more prominent than in human tumors due to significant differences in tumor structure and function [9]. Furthermore, Maeda, et al. [23] reported that the use of EPR in the imaging process based on the presence of nano-probes and nanocarriers in the tumor for a short time (a few minutes) is more appropriate than for therapeutic activities. However, they showed that the EPR effect varies based on the physicochemical properties of the nanocarriers [23]. Moreover, Danhier [17] discussed some factors that result in EPR pitfalls in clinical trials, the future of EPR, and the required changes in drug nanocarriers. Recently, Wu [22] reported that despite the meager success of the EPR effect in clinical trials, it can be enhanced by regulating the physicochemical properties of nanocarriers, treatment duration, and the type of anti-cancer agents. At the same time, Izci, et al. [24] explained the advantages and challenges of EPR, the EPR enhancement mechanisms, and the study of independent EPR-based drug delivery strategies.

Although these review articles have comprehensively discussed the EPR-based therapeutics of solid tumors, people have not surveyed EPR-based tumor targeting with a focus on fluid flow and convection/diffusion mechanisms. Therefore, this review discusses changes in structure as the tumor grows and how fluid flow, convection/diffusion, and the physicochemical properties of nanocarriers affect the EPR-based transport of nanocarriers in solid tumors. Additionally, we discuss the need for improved novelty methodologies in analyzing the EPR effect.

## 2. Effective Vascular Structures for EPR in Solid Tumors 

Solid tumors, unlike hematologic cancers, have a structure similar to normal tissue, including the parenchyma, which contains neoplastic cells and the surrounding stroma [25,26]. The stroma of solid tumors is critical for cancer cell survival and metabolism through extracellular matrix components and blood vessels [27,28]. Despite the similarity of the main components of the stromal extracellular matrix of solid tumors and normal tissue, solid tumors have different stromal patterns based on early- or late-forming blood vessels, which play an important role in therapeutic efficacy [29,30]. For example, the highest ratio of stroma to tumor cells is found in gastric and pancreatic cancer tissues [31,32], while medullary breast carcinoma and lymphomas have the lowest stromal content [27]. Hence, solid tumor therapy with one type of drug-based nanoplatform could not result in comparable clinical efficacy between these tumor types. Changes in the heterogeneity and aggressiveness of solid tumors are derived from the extent of interactions between tumor cell populations [33]. Targeting stromal cell-mediated pro-angiogenic signals in the TME is a key therapeutic strategy. The heterogeneity of angiogenesis and blood vessel maturation in human tumors can be regarded as a critical factor, resulting in conflicting outcomes in therapeutic studies. Based on the growth of solid tumors, it is possible to observe mature and immature vessels within the same tumor, resulting in different EPR effects. In fact, six groups of vessels in solid tumors can be observed with various sizes and shapes [34] (Figure 1A), which include (1) mother vessels (very large, thin-walled, permeable, with weak pericytes), (2) glomeruli microvascular (poor organization of proliferated endothelial cells, pericytes, and basement membranes), (3) capillaries (including primary and microvascular glomeruloid vessels), (4) vascular malformations (often with asymmetric coverage of smooth muscle cells and or tissue), (5) feeding arteries (large vessels with complete capillary structure and often torturous), and (6) drainage veins (very large with complete capillary structure). Late detection of solid tumors in humans, unlike in model animals in which cancer is screened in the early stages, can also increase vascular growth, maturation, and diversity, which can serve as potential barriers to drug delivery in solid tumors [24]. It is well known that biological sex and vascular development affect vascular architecture and permeability [35]. Therefore, the impairment of newly formed blood vessels (types 1 to 4) during tumor angiogenesis could not lead to improved efficacy of conventional anticancer therapies. Moreover, during blood vessel maturation in solid tumors, the vessels show a variety of complexities that affect fluid flow resistance and may even generate reversed fluid flows [36]. Additionally, the asymmetric vascular growth, vasoconstriction, and solid stress found in desmoplastic tumors create altered physical forces around the blood vessels [16,37,38] (Figure 1B), which induce low red blood cell fluidity and increased flow resistance, which can induce hypoxia and acidosis. Furthermore, angiogenesis and vascular remodeling in normal tissues follow the law of diameter reduction in smaller branches, which is not seen in solid tumors [39]. Therefore, the heterogeneity of solid tumor microvascular networks could result in alterations in vascular permeability and interstitial fluid flow, which act as key biological barriers to cancer drug delivery and efficacy in solid tumors. The overgrowth of solid tumors and a fibrotic response with increased ECM deposition lead to the introduction of mechanical barriers, resulting in altered interstitial fluid flows [40,41]. Heterogeneity of vascular fluid flow due to physical barriers due to increased cell proliferation in some solid tumor areas can reduce the efficiency of drug nanocarriers, especially in dimensions larger than 100 nm, intratumoral transport, and the EPR effect [20,41].

Regardless of the stages of vascular formation in solid tumors, which have been studied by Fang et al. [42], exploring the molecular characteristics of tumor vessels can provide a more accurate understanding of the active or inactive delivery of drug nanocarriers. Unlike normal vascular architecture with a homogeneous distribution of endothelial cells surrounded by pericytes, the heterogeneous vasculature of a solid tumor results in different permeability models [43]. This morphological abnormality not only provides greater vascular permeability in solid tumors but also leads to increased interstitial pressure in the tumor tissue, which generates multiple distribution patterns of drug nanocarriers [44]. It seems that determining the balance between vascular resistance and vascular leakage of solid tumors can be used as a basis for drug delivery mediated by the EPR effect to increase therapeutic efficacy in interstitial fluid. Basically, momentary changes in fluid flow with vascular resistance, and vascular leakage with vascular wall pore diameters ranging from 50–100 nm (tumor vessels with poor permeability) to 500–1000 nm (high permeability vessels) are explained. Overall, the EPR effect is expected to be more effective during the early phases of cancer growth or peritumors than late phases due to improved leakage behavior and the rate of fluid flow. This result could advance future therapeutics for solid tumors with a different approach.

## 3. EPR-Mediated Drug Delivery to Solid Tumors 

In recent years, various perspectives on anticancer nanomedicine have been developed based on nanocarrier intratumoral transport pathways [22,45,46,47]. The structural complexities of solid tumors that affect drug delivery pathways remain unclear [15]. As such, the blood flow complexities mentioned in Section 2, such as obstruction of fluid flow in the deeper parts of the tumor, tumor heterogeneity due to different types of vessels that cause interstitial fluid pressure, diversity of ECM of solid tumors, and lack of lymphatic drainage vessels, have caused a variety of therapeutic challenges based on the intratumoral transport and drug efficacy [20,25,48]. Regardless of the method and location of the injection, the drug nanocarriers introduced in vivo are distributed intratumorally using two basic concepts. Convection is the transfer of drugs by a moving fluid, such as blood or interstitial fluid, in which case the distance between the fluid and the cancer cell is critical [49], and diffusion is proportional to the concentration gradient, which is effective at short intervals of drug injection) [50]. Meanwhile, the intratumoral transport of drug nanocarriers in solid tumors is a function of both phenomena, based on fluid flow in the solid tumor, which strongly depends on the tumor size [51]. In addition, diffusion is more prevalent in the case of smaller-sized drug nanocarriers in small vessels as well as in the intercellular spaces [52,53] similar to the pulmonary tissue vessels [54]. However, convection is more prevalent in the case of larger drug nanocarriers present in a solid tumor core [55] and large vessels [53,56]. In this regard, it has been determined that the specific entry of therapeutic payloads into tumors is inhibited when diffusion is involved in the intratumoral transport of nanocarriers due to the increase of interstitial pressure in solid tumors [57,58,59]. Hence, diffusion can sometimes be considered an obstacle to yielding novel, effective therapies for solid tumors. Furthermore, if the interstitial fluid pressure stops as a function of tumor solid stress [60], or fluid flow in small vessels is reduced [61], EPR-mediated drug delivery could fail. Thus, the changes in fluid velocity in different parts of solid tumors based on their heterogeneous development and progression can be considered a very important indicator of the efficacy of EPR-mediated drug delivery in solid tumors. As the role of interstitial fluid pressure [59] and tumor solid stress [62] in modulating fluid velocity is very important, it is expected that the intratumoral transport of nanocarriers and the EPR effect in the peritumoral areas will be greater than that of the core of solid tumors. In addition, the change in the direction of fluid flow in solid tumors [63,64] due to the irregular structure of vessels can result in the mitigation of convection/diffusion-enhanced delivery of drugs for the treatment of solid tumors. By reversing the flow, the viscosity of the fluid in solid tumors is expected to be increased several times that of normal tissue due to an increase in hematocrit [65]. Therefore, this phenomenon can effectively manipulate convection and diffusion events in interstitial transport in solid tumors mediated by EPR. 

Tumor vessel development and maturation can also lead to the formation of an integrated structure of endothelial cells and the presence of pericytes, which changes the mechanism of intratumoral transport of drug nanocarriers due to an alteration in the ratios of convection/diffusion status [58,64]. Although reducing the number and size of vascular wall pores moderates the passive transport of drug nanocarriers into the interstitial space, computational models predict that even without vessel pores/gaps due to decreased interstitial fluid retention and reduced leakage into vessel lumen with higher uptake through intercellular and linked vesicles effectively increases the tumor delivery of macromolecular drugs based on the EPR effect (Figure 2) [64]. 

However, due to capillary growth and the reduction of leaky regions of the tumor vasculature, the passive transport of drug nanocarriers is significantly reduced. Increasing the thickness and density between the vascular lumen and the interstitial space due to vascular maturation along with greater ECM and collagen accumulation can dramatically reduce the potential tumor-targeted drug delivery based on EPR-effect [66]. Therefore, an updated view of the transport of drug nanocarriers based on the EPR effect due to the heterogeneous vasculature of a solid tumor is needed.

## 4. Physicochemical Properties of Nanoscales Affecting EPR 

Numerous reports have indicated the effect of the size, shape, and surface charge of drug nanocarriers on the transport of drugs based on the EPR effect in cancer [67,68,69,70]. However, most accumulation, retention, and drug transfer kinetics in solid tumors have been investigated in animal models without accessing human data. Therefore, despite various reports on the therapeutic advantages of drug nanocarriers for the treatment of tumors (Table 1), significant advances have still not been observed in clinical trials. Of course, the use of other treatment modalities, such as photothermal therapy [71], photodynamic therapy [72], magnetic field therapy [73], sonodynamic therapy [74], has been able to boost the efficacy of drug nanocarriers in tumor therapy. However, the heterogeneity of solid tumors and the effect on EPR-mediated passive drug targeting result in a non-uniform distribution of the drug in solid tumors [75], which increases the likelihood of local cancer recurrence.

### 4.1. Size of Drug Nanocarriers

Drug delivery through the EPR effect is related to the biocompatibility of drugs and their molecular size to escape renal clearance. Reports indicate that drugs with molecular weights of over 40 kDa can increase the EPR effect on solid tumors and other organs by increasing their shelf life in the blood. Therefore, in order to increase the shelf life of drugs less than 40 kDa, the combination of drugs with polymers or serum proteins, such as albumin, is recommended [7]. Nevertheless, despite the direct and uniform effect of drug molecular size and drug nanocarrier size on EPR via increasing shelf life more than a few hours, nanocarrier size directly affects EPR by changing their transport mechanisms to the interstitial space based on fluid velocity, and inducing accumulation in the tumor site by reducing leakage into the vessel lumen [7,81]. If the fluctuations in fluid flow in the vessels are minimized, small-sized drug nanocarriers (20–70 nm) tend to penetrate further into solid tumor cores compared to larger ones [67,82]. Thus, the size of the spherical nanocarriers plays a key role in EPR-based intracellular targeted delivery of drugs. Therefore, the transport of smaller-sized and larger-sized spherical nanocarriers becomes a function of convection and diffusion, respectively [53]. Based on this fact, in the case of larger nanocarriers, it is necessary to accelerate the fluid velocity to overcome interstitial pressure, which prevents deep penetration of nanocarriers into tumor tissue [83]. However, by using the ligand-receptor binding approach in targeting nanocarriers, the effect of nanoparticle size and shape on EPR-based drug delivery can be moderated by enhancing the penetration of nanocarriers into tumors without affecting fluid flow [24]. Hence, the success of EPR-based nanocarriers in the interstitial space is strongly dependent on the size of the nano-based platform and interstitial fluid flow velocity. In addition to vascular flows, the dense matrix in the interstitial space of tumors [84] dramatically reduced the penetration of larger-sized drug nanocarriers (150–200 nm) compared to smaller-sized drug counterparts (20–70 nm) [85]. In addition, drug nanocarriers below 10 nm are not potential candidates for efficient tumor-targeted drug delivery based on the EPR effect in solid tumors due to rapid clearance by kidneys and macrophages located in the liver, lung, and pancreas [86]. 

As a result, to improve the EPR effect in solid tumors to predict the therapeutic efficacy of nano-based drugs, the lack of rapid clearance, increased permeability in solid tumors, and the ability to use combined therapeutic modalities should be considered.

### 4.2. Shape and Surface Charge of the Drug Nanocarriers

Other factors, including the shape and surface charge of drug nanocarriers based on changing the nature of transport, improving shelf life, and intracellular permeability, can affect the EPR-based therapeutics of solid tumors. It has been observed that non-spherical nanocarriers can have higher diffusivity than spherical nanocarriers as a result of increased interactions with the vessel wall through improved radial thrust force deriving from rapid pressure changes [7,87,88]. Thus, the fluid provided a higher radial thrust force by applying multilateral force to different surfaces of non-spherical nanocarriers (Figure 3B). In contrast, spherical drug nanocarriers show convection-enhanced delivery of targeted drug penetration into solid tumors (Figure 3A) [89]. Based on radial thrust, Chauhan, et al. [90] reported that rod nanocarriers were retained 4 times higher than spherical nanocarriers in solid tumors (Figure 3B). In addition, in a 3D spheroid model, it was recognized that cylindrical nanocarriers with 100 nm height and 325 nm diameter have maximum delivery to solid tumors and a more uniform penetration pattern than their nanorod counterparts, while much smaller non-spherical nanocarriers penetrated deeper and more uniformly compared to larger non-spherical nanocarriers [91]. Therefore, the deformation of nanocarriers based on diffusion and convection results in heterogeneous distributions of nanomedicines in solid tumors. However, the combination of the size and shape factors of drug nanocarriers can exhibit different responses based on the various behaviors of drug nanocarriers with respect to interstitial fluid flow in tumors. For instance, it has been reported that spherical drug nanocarriers with dimensions less than 70 nm have a higher radial thrust force than spherical nanocarriers with dimensions greater than 130 nm [92]. Furthermore, non-spherical nanocarriers have a longer half-life due to less uptake by macrophages, which can increase the probability of deep penetration into the tumor [93]. The non-spherical shape of nanocarriers also improves drug-mediated endocytosis due to increased interactions between endothelial cells and drug nanocarriers [88]. Together, this will result in a more effective penetration of anticancer drugs through tumor tissue. However, due to the negative charge of vessels [94], EPR-based drug delivery by nanocarriers can be manipulated by using a positive charge on nanocarriers through modulation of radial thrust [95] based on electrostatic interactions. For example, Campbell, et al. [96] and Krasnici, et al. [97] showed that liposomal nanocarriers with dimensions of ~150 nm and positively charged surface area could accumulate 1.5 and 7 times higher than that of anionic and highly anionic liposomal nanocarriers in the solid tumor, respectively. However, their results showed that cationic liposomes did not potentially penetrate the intratumoral space compared to the other two samples. Therefore, the possibility of low penetration of drug nanocarriers in the interstitial space due to the potential interaction of positively charged nanocarriers with the vascular wall is also conceivable [98]. Nevertheless, the application of a positive charge faces serious challenges due to its high toxicity to non-target tissues and rapid clearance through opsonization [99,100]. The use of chemical surface modification approaches for drug nanocarriers, such as the use of compounds that are positively charged in the acidic environment of the tumor [101], can improve the surface charge drawbacks of nanocarriers.

## 5. Challenges and Common Approaches to EPR Analysis 

To achieve improved EPR-based drug delivery in solid tumors by nanocarriers, strategies such as the development of potential drug nanocarriers, injection time periods of drug nanocarriers, imaging of solid tumors, evaluation of pharmacokinetic and pharmacodynamic parameters, and qualitative and quantitative analysis of cancer specimens should be considered (Figure 4). The most common imaging techniques used during the treatment process are MRI, SPECT/CT and PET/CT imaging techniques in humans and sometimes in animal models and fluorescence imaging in in vitro models. Staining methods for microscopic observations and electron microscopy in qualitative techniques and atomic absorption, magnetic separation, flow cytometry, chromatography, and centrifuges in quantitative methods are the most common analytical strategies. The inconsistency of evaluation methods between human and animal models and between in vitro and in vivo methods has made it difficult and sometimes impossible to compare outcomes.

By reviewing the literature, we can see that the main challenge in EPR-based trial approaches is the lack of standardization of animal models used in clinical assays for humans. For instance, humans simultaneously face challenges such as metabolic problems (obesity, diabetes, stress, high cholesterol or blood pressure, and age-related vascular problems), an unhealthy diet, and stressful lifestyles that are not present in animal models. Additionally, the different sizes of animals, especially rodents compared to humans, alter the pharmacokinetic and pharmacodynamic processes, which is often not considered. The resulting distribution, stability, and retention of drug nanocarriers are closely related to the above parameters. Moreover, tumor induction in mice with intensified growth patterns and extraordinarily large sizes relative to body mass compared to humans can alter EPR results by altering tumor cell population, tumor spatial structures, and intrinsic difference in fluid flow [102]. The rate of blood flow in mouse tissues is approximately 800 times lower than that of humans. For example, the blood flow in mouse C3 and human lymphosarcoma tumors is 5.4 mL/100 g per min [103] and 40–64 mL/100 g per min [104], respectively. Since tumor blood flow affects the penetration of nanocarriers and the delivery of EPR-based antitumor drugs based on the slower blood flow rate in mice than in humans, a more appropriate situation for the treatment of mouse tumors is conceivable. Nonetheless, in many research activities to investigate the EPR effect in solid tumors, this factor has been neglected, and the best solution is to use animal models with more similar body weights as humans.

The second error that should be investigated is the lack of standardized information on the injection times of drug nanocarriers and duration of therapy, i.e., the treatment schedule, without adopting it to human conditions. Although the use of drug nanocarriers is modified based on the weight to match the effects of the drug in mice and humans, the injection periods and duration of treatment used in research activities generally depend on the durability of the drug and tumor size. These factors result in fundamental differences in the rate of clearance of drug nanocarriers between humans and animals based on fluid velocity, clearance, and solid tumor size. Therefore, due to the different stability of drug nanocarriers in different species, it is expected that the process of evaluating EPR effects in animal and human solid tumors will differ.

Another important challenge is the lack of evaluation of the net efficacy of the EPR effect on the intratumoral accumulation of drug nanocarriers. EPR-based assays are generally performed based on the administration of a specific dose of drug nanocarriers intravenously or orally and the analysis of subsequent retention in tumor tissues. Based on the clearance challenges mentioned in the previous sections, it seems that the administration of drug nanocarriers through tumor-associated arteries can truly reveal the net retention efficiency in solid tumors. For this strategy, the direct injection of drug nanocarriers into the tumor-associated arteries by imaging-guided catheter placement can be used. Explaining the net efficacy of drug nanocarriers in increasing EPR could influence the opinion of researchers to use a variety of drug nanocarriers to treat solid tumors by modifying the production process. Nevertheless, few drug nanocarriers have been reported for arterial direct injection into solid tumors [105,106].

Other obvious errors in EPR detection processes include the method of collecting samples based on the structural status of solid tumors and the accuracy of sample preparation. As mentioned above, the penetration efficacy of drug nanocarriers in the solid tumor core is lower than that of the peritumoral region. Thus, it is more desirable in therapeutic approaches to make a detailed distinction between different areas of the tumor in the EPR examination to explain the effect of drug nanocarriers on therapeutic pathways in a more accurate and reproducible way. In addition, it seems that complete blood withdrawal should be performed to prevent the report of drug nanocarriers in the blood, as nanocarriers are retained from solid tumor samples.

Although the mentioned challenges in controlling the concentration of drug nanocarriers in solid tumors of animal models can be relaxed due to ethical sensitivities, possible toxicities, high cost, and the complexity of experiments, the use of the above methods in clinical practice faces serious challenges.

## 6. Convergence of Theories to Reduce Conceptual Shortcomings in EPR

For almost 45 years, EPR theory has been pursued by researchers as an important and reliable effect in cancer therapy. However, in recent years, a group of researchers have hypothesized the ineffectiveness of the EPR effect in cancer therapy [17]. They believe that the different outcomes in cancer treatment derived from animal and human models are a challenge to EPR effect-based cancer therapeutics due to differences in body structure and solid tumor morphologies. At first glance, based on Section 2 and Section 3, this hypothesis is logical. However, could the potential pitfall for the improvement of tumor uptake by nanocarriers be considered a criterion for accepting/rejecting the EPR effect hypothesis? Could the previous results of EPR effect-based tumor targeting and cancer nanomedicine treatment efficacy be advanced with further strategies? The results of several studies demonstrate that the retention of drug nanocarriers by increasing the imbalance in the inflow (through vascular wall disorders) and fluid output (due to defects in the tumor lymphatic system) increases the killing of tumor cells in solid tumors [9,107]. Although the lack of reliable human data has challenged the EPR effect-based anticancer nanomedicine drug targeting, assuming that clinical outcomes are always a function of solid tumor heterogeneity [108], physicochemical features of drug nanocarriers [109], and therapeutic designs with combined modalities (photo-therapy, thermal-therapy, magnetic-therapy, etc.) [24], it is not possible to precisely attribute the rates of treatment failure to one of the existing therapeutic factors. Nevertheless, the convergence of the different theories can serve as a platform for optimizing cancer treatment strategies. 

The first new cancer treatment strategy based on the EPR effect is to change the perspective of EPR effect-based therapeutic approaches. In the treatment process, the transfer of drug nanocarriers from the cancer stroma to malignant cells is essential [110]. As seen in several papers, the high accumulation of drug nanocarriers in normal tissues can lead to undesirable toxicity through drug release and catalytic activity [111]. With this new perspective, it can be hoped that the actual accumulation of drug nanocarriers into solid tumors will result in a promising therapeutic target for cancer treatment, especially by enhanced tumor penetration of the drug. Therefore, to enhance therapeutic activities in solid tumors, accepting the EPR effect, along with recruiting combined therapeutic modalities to trigger the penetration of drug nanocarriers into malignant cells, can result in improved nano-based precision carriers for targeting therapeutic and diagnostic agents in the treatment of cancer. However, the accumulation of drug nanocarriers in solid tumors, especially in peritumoral areas, could be an EPR-independent strategy. Therefore, based on the nature of the nanocarriers and drug availability, a new delivery nanoplatform could be developed for tracking and treating poorly vascularized solid tumors.

Another strategy to modulate EPR is the manipulation of drug nanocarriers through surface charge modifications, cluster size, flexible shape, and surface coatings for greater stability, with or without external inducers, to augment the permeability of tumor-associated vessels [6,112,113]. Modifications applied to drug nanocarriers to enhance the EPR effect usually do not result in potential active targeting. Since the effect of EPR indicates a high accumulation of targeted nanocarriers in the stromal tissue of solid tumors, overtaking the inflow on outflow of drug nanocarriers can lead to inactive or semi-active accumulation of nanocarriers in solid tumors using the EPR effect. 

Competition is also one of the most important strategies in the development of therapeutic nanomedicine for solid tumors. Indeed, the main competitors of EPR-based treatment of solid tumors are the liver, spleen, lung, and kidney. Therefore, reducing the clearance of drug nanocarriers by modulating the physicochemical properties of nanocarriers matched to the TME of solid tumors can improve EPR-based tumor targeting and cancer nanomedicine treatment efficacy. As mentioned in Section 3, altering the physicochemical properties of drug nanocarriers can alter penetration into tumor tissue. For example, binding molecules or coatings of drug nanocarriers can modulate the renal filtration rate [114], or natural and synthetic polymers can be used to minimize uptake by the liver [115]. On the other hand, the different morphological structures of solid tumors are key factors in designing and developing potential targeting drug nanocarriers. This means that EPR effect-based drug delivery is much more effective in primary tumors with vascular structures 1 to 4 than in mature tumors with vascular structures 5 to 6 [116]. Therefore, quantifying the morphological structures of solid tumors by various methods, such as imaging and histopathology, can highlight the ability of the EPR effect to improve the efficacy of anticancer nanomedicine. For instance, it has been determined that in micro-metastatic tumor cell clusters the possibility of EPR effect-based drug delivery is much lower than that of other tumors [117].

Standardization of the methodology for analyzing the EPR effect in solid tumors can enable researchers to develop the function of EPR effect-based treatment of solid tumors. According to Section 2 and Section 5, exploring the morphology of solid tumors and their development can provide useful information about the EPR effect-based tumor delivery of nanodrugs, intracellular infiltration, cancer cell death, and changes in the physiological behavior of the tumor. It should be noted that the main cause of EPR effect-based dependent or independent treatment of solid tumors is due to the intratumoral morphological heterogeneity of solid tumors and the different methodologies used in EPR effect analysis. Therefore, due to the presence of different methodologies, along with the variable structure of the solid tumor, reports on the EPR effect-based treatment of tumors are conflicting.

The timetable of treatment interventions is a critical challenge that can limit the EPR effect on the treatment of solid tumors. The review of Park, Choi, Chang, Um, Ryu, and Kwon [46] confirms the alliance of the EPR effect and the reduction of hypoxia levels in solid tumors based on greater oxygen permeability using drug nanocarriers [118]. Reducing the level of hypoxia by the EPR effect increases the possibility of the potential treatment of solid tumors [119,120]. However, different responses to solid tumors have been observed with increasing oxygen levels [121]. Furthermore, due to the lack of sufficient information on the relationship between physiological changes in solid tumors with decreasing hypoxia and metastatic status, the role of the EPR effect in macromolecular therapeutics is still unknown in detail [122,123]. 

## 7. Conclusions

Although therapeutic strategies used in solid tumors based on the EPR effect in animal activities are more successful than in human activities, a successful EPR-based treatment process can be established by increasing the structural insight of solid tumors in patients, changing the methods of manufacturing drug nanocarriers, and improving EPR analysis approaches. Finally, we emphasize that addressing one-dimensional issues rather than multidimensional concepts eliminates the possibility of integrating factors that affect the treatment of solid tumors. This eventually leads to more expensive and sophisticated approaches that divert public access from promising treatments. The current effort is to overcome the misconceptions of traditional EPR-related nanomedicine with the convergence of different ideas.

## Figures and Tables

**Figure 1 cancers-14-02868-f001:**
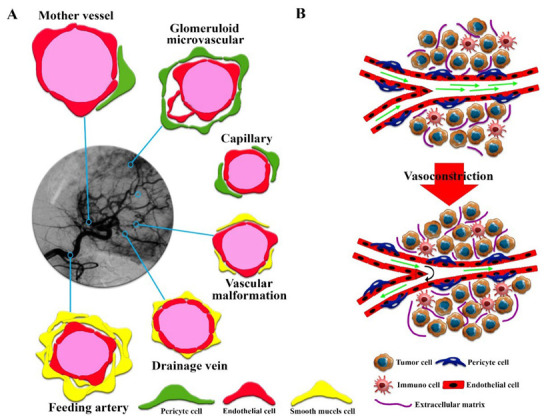
(**A**) Abnormally patterned vascular vessels in solid tumors. Six types of blood vessels with different characteristics can be identified. (**B**) A description of the overall structure of solid tumors and the solid stress phenomenon caused by tumor tissue growth that reduces fluid flow and even reverses fluid flow in the tumor.

**Figure 2 cancers-14-02868-f002:**
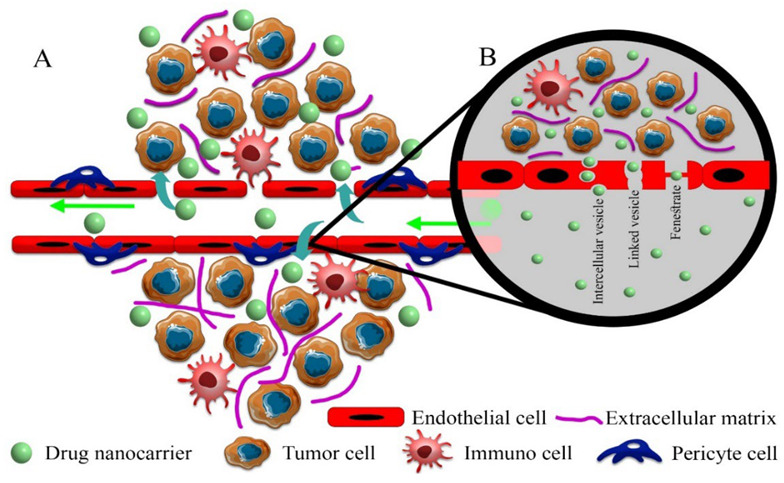
Proposed pathways for drug nanocarriers to enter solid tumors. (**A**) Paracellular process: in this pathway, drug nanocarriers passively enter the extracellular space of solid tumors through intercellular gaps with dimensions up to 2000 nm, which are very important in the EPR effect. (**B**) Transcellular process: drug nanocarriers in mature vessels without common gaps in solid tumors actively enter the extracellular space of the solid tumor through vesicles (endocytosis-exocytosis) and pores. The fenestrate pathway does not have a complete incision, and a diaphragm separates the internal space from the lumen of the vessel. In the linked vesicle path, interconnected vesicles cause the transfer of drug nanocarriers.

**Figure 3 cancers-14-02868-f003:**
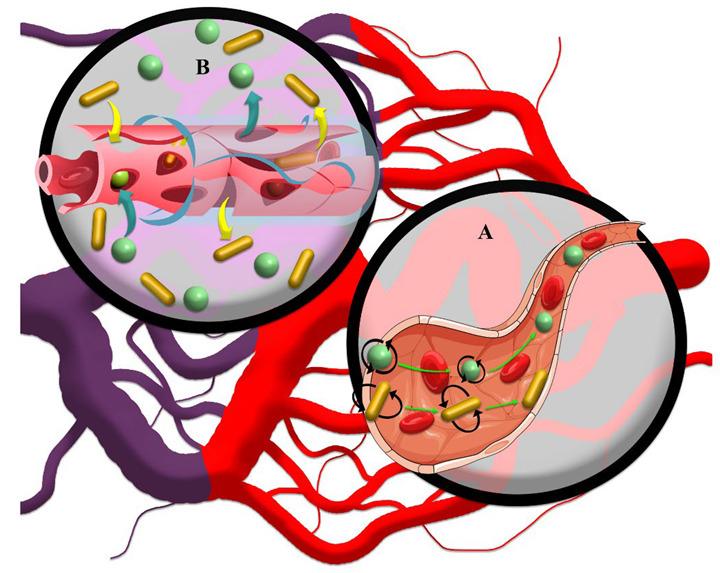
(**A**) Regardless of size, spherical drug nanocarriers tend to penetrate the solid tumor core based on the convection current. The forces exerted on the moving rod-like drug nanocarriers cause them to marginalize and further attach to the vessel wall and incomplete vessel structure, which has many gaps for leakage of drug nanocarriers. (**B**) Based on high interstitial pressure in solid tumors, rod-like drug nanocarriers show a further EPR effect in the interstitial space than spherical drug nanocarriers due to their lower tendency to move.

**Figure 4 cancers-14-02868-f004:**
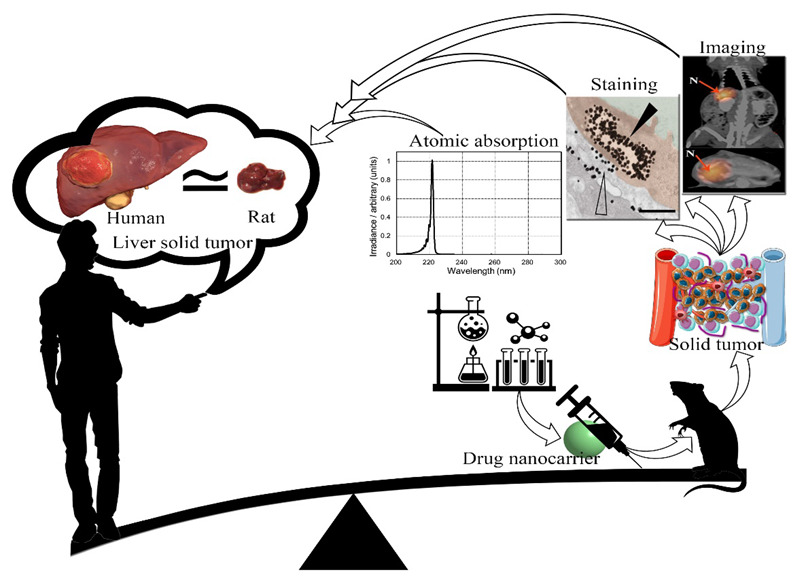
Schematic view of the production process of drug nanocarriers for the treatment of solid tumors in animal models and their generalization to humans. It seems that, due to dissatisfaction with the research achievements in the treatment of solid tumors mediated by an EPR effect, it is necessary to update the experimental strategies in this field.

**Table 1 cancers-14-02868-t001:** Summary of in vivo pharmacokinetic (PK) studies of conventional chemotherapeutics (CC) and targeted anticancer agents (TAA).

Drug (Nanocarriers)	Outcomes: TAA vs. CC	Cellular/Tumor Uptake	Ref.
Docetaxel (TMCC-co-LA)-g-PEG	Raised AUC (2 folds); reduced V_d_ (2 folds); prolonged t_1/2_ (1.6 folds); decreased CL (3 folds)	2-fold increase in drug concentration within 8 h and 5-fold decrease in drug clearance from tumor	[76]
Doxorubicin (PAD–PPI)	Increased AUC (3.2 folds); decreased CL (3.12 folds).	4-fold raise in drug concentration within 8 h	[77]
Docetaxel (PLGA–mPEG)	Improved AUC (2.7 folds), prolonged t_1/2_ (3.76 folds), reduced CL (2.7 folds)	3.5-fold increase in drug concentration within 16 h	[78]
Doxorubicin (Mannosylated- SLNs)	Raised AUC (5 folds); prolonged t_1/2_ (9.3 folds); decreased CL	2.8-fold raise in drug concentration within 8 h and 2.4-fold decrease in drug clearance from the tumor	[79]
Docetaxel (CMC–PEG)	Increased AUC (38.6 folds); prolonged t_1/2_ (5.2 folds); decreased CL (2.5%); decreased V_d_ (13.2%)	Tumor uptake was 5.5-fold more than that by free drug within 3 h and 2.5-fold decrease in drug clearance from tumor	[80]

AUC, area under plasma concentration–time profile; CL, total clearance; CMS-PEG, PEGylated carboxymethyl cellulose; mPEG-PLGA, methoxy poly (ethylene glycol)-b- poly (lactic-co-glycolic acid); PAD-PPI, polyaldehydodextran-polypropylene imine; SLN, solid lipid nanoparticles; t_1/2_, elimination half-life; TMCC-co-LA-g-PEG, poly (2-methyl-2-carboxytrimethylene carbonate-co-D,L-lactide)-graft-poly(ethylene glycol); V_d_, apparent volume of distribution.

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
