# Peer review of "An Updated Review on EPR-Based Solid Tumor Targeting Nanocarriers for Cancer Treatment"

_cancers, 2022, doi:10.3390/cancers14122868_

Round 1

Reviewer 1 Report

The autors made a review on EPR-based solid tumor targeting nanocarriers for cancer treatment.

It is an extensive work that is very well founded. It is a very precious work for the researchers on this field.

It is well conducted and the bibliography is quite good.

Author Response

Thank you

Reviewer 2 Report

Dear Authors 

The review “An updated review on EPR-based solid tumor targeting nanocarriers for cancer treatment” is well addressed and structured in the field of the development of novelty alternatives for cancer treatment.

This contribution is well written, minor corrections are attached in the word file.

Please improve the references of the manuscript, excessive self-citations were found.

Best regards.

Author Response

The review “An updated review on EPR-based solid tumor targeting nanocarriers for cancer treatment” is well addressed and structured in the field of the development of novelty alternatives for cancer treatment.

This contribution is well written, minor corrections are attached in the word file.

Based on recommendation, spelling and grammatical errors were corrected in the paper.

DONE

Please improve the references of the manuscript, excessive self-citations were found.

Based on recommendation, 6 references deleted and new references such as [1], [35] and [73] replaced.

DONE

Reviewer 3 Report

The authors in this review discuss valuable insights into the limitations of the EPR effect in therapeutic efficacy and report crucial perspectives on how the EPR effect can be modulated to improve the therapeutic effects of nanomedicine. The manuscript to me is, in general, clearly written. The science and technical execution of the manuscript is of good quality. The manuscript is solid and the data, in general, support the conclusions. The theory, logic, and experimental design are easy to follow and in general make sense.

Specific comments

- The title should not contain abbreviations.

 - Line 21, 22: check the font size.

- Line 36: delete the last ( ; )

- Introduction is a bit long for review articles, try to shink it by 50 %.

- Line 55, 60: there is a mise use of abbreviations.

_ There are many long sentences in Lines 58-63, 150-155, and so on. rephrase it.

- Line 86: delete the word will.

- Lines 109-116: the authors described several kinds of veins and only one kind of artery ! is this correct? also, how about the center of solid tumors where there are almost no blood vessels?

- How about the EPR in naturally resistant solid tumors?

_ is there a difference in EPR of conventional chemotherapeutics and targetted anticancer agents?

_ In section 4, add some pharmacokinetics data about the intratumor concentrations of EPR-nanocarriers compared to normal anticancer drugs.

- I encourage the authors to add a table summarizing sections 3,4, and 5.

- Lines 239, 259: 20-70 or 30-60?

- Others are ok.

Author Response

The authors in this review discuss valuable insights into the limitations of the EPR effect in therapeutic efficacy and report crucial perspectives on how the EPR effect can be modulated to improve the therapeutic effects of nanomedicine. The manuscript to me is, in general, clearly written. The science and technical execution of the manuscript is of good quality. The manuscript is solid and the data, in general, support the conclusions. The theory, logic, and experimental design are easy to follow and in general make sense.

Specific comments

- The title should not contain abbreviations.

Based on the word limit by the journal and as EPR is a well-known abbreviation, we would prefer to not reword the title

 - Line 21, 22: check the font size.

Based on recommendation, the font size was corrected in line 21 and 22.

DONE

- Line 36: delete the last ( ; )

Based on recommendation, the ; was corrected to . in line 35.

DONE

- Introduction is a bit long for review articles, try to shrink it by 50 %.

The authors believe that the length of the introduction was appropriate (less than 8% of the total article) and described a suitable initial topic for the paper.

- Line 55, 60: there is a misuse of abbreviations.

Based on recommendation, the TME was corrected to TIME in line 55 and 60.

DONE

_ There are many long sentences in Lines 58-63, 150-155, and so on. Rephrase it.

Based on recommendation, the long sentences were rephrased in lines 58-62 and 150-155.

DONE

- Line 86: delete the word will.

Based on recommendation, the word was deleted in line 86.

DONE

- Lines 109-116: the authors described several kinds of veins and only one kind of artery! is this correct?

The authors state that the classification of blood vessels in the 109-116 lines is correct based on various reports (https://doi.org/10.1038/s41598-020-77180-1, doi: 10.2147/HP.S133231, https://doi.org/10.1371/journal.pone.0010282). However, according to published reports, there is a feeding artery and a drainage vein that are in the classification (figure 1). Other vessels, depending on their condition, can be arteries or veins, for which the term microvascular is used, and their structure is roughly described.

_ Is there a difference in EPR of conventional chemotherapeutics and targeted anticancer agents?

Based on recommendation, the difference in EPR of conventional chemotherapeutics and targeted anticancer agents was prepared in line 246-256.

DONE

_ In section 4, add some pharmacokinetics data about the intratumoral concentrations of EPR-nanocarriers compared to normal anticancer drugs.

Based on recommendation, the pharmacokinetics data were prepared in table 1.

DONE

- I encourage the authors to add a table summarizing sections 3, 4, and 5.

Thanks for the proper attention of the referee. The authors believe that the reports presented in Sections 3, 4 and 5 are sufficiently informative and that tables will be used in future research.

- Lines 239, 259: 20-70 or 30-60?

Based on recommendation, the number was corrected to 20-70 nm in line 239 and 259.

DONE

Round 2

Reviewer 3 Report

The manuscript is substantially improved and could be published in its current form.